# Identification of four novel associations for B-cell acute lymphoblastic leukaemia risk

Jayaram Vijayakrishnan [1,19], Maoxiang Qian[2,3,19], James B. Studd [1], Wenjian Yang[2], Ben Kinnersley [1], Philip J. Law [1], Peter Broderick [1], Elizabeth A. Raetz[4], James Allan[5], Ching-Hon Pui [6,7], Ajay Vora[8], William E. Evans [2,7], Anthony Moorman[9], Allen Yeoh[10,11], Wentao Yang[2], Chunliang Li [12], Claus R. Bartram[13], Charles G. Mullighan [6,7,14], Martin Zimmerman[15], Stephen P. Hunger[16], Martin Schrappe[17], Mary V. Relling[2,7], Martin Stanulla[15], Mignon L. Loh[18], Richard S. Houlston [1]* & Jun J. Yang [2,6,7]*

There is increasing evidence for a strong inherited genetic basis of susceptibility to acute lymphoblastic leukaemia (ALL) in children. To identify new risk variants for B-cell ALL (B-ALL) we conducted a meta-analysis with four GWAS (genome-wide association studies), totalling 5321 cases and 16,666 controls of European descent. We herein describe novel risk loci for B-ALL at 9q21.31 (rs76925697, $P = 2.11 \times 10^{-8}$), for high-hyperdiploid ALL at 5q31.1 (rs886285, $P = 1.56 \times 10^{-8}$) and 6p21.31 (rs210143 in *BAK1*, $P = 2.21 \times 10^{-8}$), and *ETV6-RUNX1* ALL at 17q21.32 (rs10853104 in *IGF2BP1*, $P = 1.82 \times 10^{-8}$). Particularly notable are the pleiotropic effects of the *BAK1* variant on multiple haematological malignancies and specific effects of *IGF2BP1* on *ETV6-RUNX1* ALL evidenced by both germline and somatic genomic analyses. Integration of GWAS signals with transcriptomic/epigenomic profiling and 3D chromatin interaction data for these leukaemia risk loci suggests deregulation of B-cell development and the cell cycle as central mechanisms governing genetic susceptibility to ALL.

[1] Division of Genetics and Epidemiology, The Institute of Cancer Research, 15 Cotswold Road, Sutton, Surrey SM2 5NG, UK. [2] Department of Pharmaceutical Sciences, St. Jude Children's Research Hospital, Memphis, Tennessee, USA. [3] Children's Hospital and Institutes of Biomedical Sciences, Fudan University, Shanghai, China. [4] Division of Pediatric Hematology and Oncology, New York University Langone Health, New York, New York, USA. [5] Northern Institute for Cancer Research, Newcastle University, Newcastle upon Tyne NE2 4HH, UK. [6] Department of Oncology, St. Jude Children's Research Hospital, Memphis, Tennessee, USA. [7] Hematological Malignancies Program, St. Jude Children's Research Hospital, Memphis, Tennessee, USA. [8] Great Ormond Hospital, London, UK. [9] Wolfson Childhood Cancer Research Centre, Northern Institute for Cancer Research, Newcastle University, Newcastle upon Tyne NE1 7RU, UK. [10] Centre for Translational Research in Acute Leukaemia, Department of Paediatrics, Yong Loo Lin School of Medicine, National University of Singapore, Singapore, Singapore. [11] VIVA–University Children's Cancer Centre, Khoo Teck Puat–National University Children's Medical Institute, National University Hospital, National University Health System, Singapore, Singapore. [12] Department of Tumor Cell Biology, St. Jude Children's Research Hospital, Memphis, Tennessee, USA. [13] Institute of Human Genetics, University Hospital, Heidelberg, Germany. [14] Department of Pathology, St. Jude Children's Research Hospital, Memphis, Tennessee, USA. [15] Department of Paediatric Haematology and Oncology, Hannover Medical School, 30625 Hannover, Germany. [16] Department of Paediatrics and Centre for Childhood Cancer Research, Children's Hospital of Philadelphia and the Perelman School of Medicine at The University of Pennsylvania, Philadelphia, Pennsylvania, USA. [17] Department of Paediatrics, University Medical Centre Schleswig-Holstein, Kiel, Germany. [18] Department of Pediatrics, Benioff Children's Hospital and the Helen Diller Family Comprehensive Cancer Center, University of California San Francisco, San Francisco, California, USA. [19]These authors contributed equally: Jayaram Vijayakrishnan, Maoxiang Qian. *email: richard.houlston@icr.ac.uk; Jun.Yang@stjude.org

Acute lymphoblastic leukaemia (ALL) is the most common paediatric cancer with B-cell precursor ALL (B-ALL) accounting for ~85% of the cases[1]. Although the peak age of diagnosis of ALL is between ages 2 and 5 years, some initiating somatic genomic abnormalities (e.g., chromosomal translocations) can be detectable at birth[2,3]. Both the absence of specific environmental risk factors and early onset suggest a strong inherited genetic basis for susceptibility[4–6]. Our understanding of ALL susceptibility has been informed by genome-wide association studies (GWAS) identifying 11 regions harbouring risk variants: 7p12.2 (IKZF1), 8q24.21, 9p21.3 (CDKN2A/B), 10p12.2 (PIP4K2A), 10q26.13 (LHPP), 12q23.1 (ELK3), 10p14 (GATA3), 10q21.2 (ARID5B), 14q11.2 (CEBPE), 16p13.3 (USP7) and 21q22.2 (ERG)[7–16]. ALL is a biologically heterogeneous disease with subtypes defined by recurrent initiating genetic abnormalities. After initiation, however, leukaemia cells acquire a constellation of secondary lesions. The two most common subtypes of B-ALL are ETV6-RUNX1 fusion positive and high-hyperdiploid (HD) ALL[17], each accounting for 20–25% of cases. HD ALL is characterised by a chromosome number > 51 due to the non-random gain of specific chromosomes. Subtype-specific GWAS associations have so far been identified at 10q21.2 (ARID5B) associated with HD ALL, 10p14 (GATA3) for Philadelphia chromosome-like ALL, and 2q22.3 associated with ETV6-RUNX1-positive ALL[7,9,12,18,19].

To gain a more comprehensive insight into susceptibility to ALL, we performed a meta-analysis of four GWAS from the North America[13,18,20] and Europe[7,9,12], with additional replication. We report both the discovery of four new susceptibility regions for ALL and refined risk estimates for the previously reported loci. In addition, we have investigated the gene regulatory mechanisms underlying the genetic associations observed at these risk loci by integrating genome-wide chromosome conformation capture (Hi-C) data and chromatin immunoprecipitation-sequencing (ChIP-seq), epigenomic and transcriptomic profiling to pinpoint target genes.

## Results

### GWAS meta-analysis and replication.
We conducted a meta-analysis of four GWAS B-ALL datasets: UK GWAS I, German GWAS, UK GWAS II and the COG_SJ GWAS[7,9,12,13,18,20], totalling 5321 cases and 16,666 controls of European descent. Following established quality-control measures for each GWAS dataset (Supplementary Fig. 1), the genotypes of ~10 million single-nucleotide polymorphisms (SNPs) in each study were imputed. After filtering out SNPs on the basis of minor allele frequency (MAF) and imputation quality, we assessed associations between ALL status and SNP genotype in each study using logistic regression. Risk estimates were combined through an inverse-variance-weighted fixed-effects meta-analysis[21,22]. Quantile–quantile (Q–Q) plots for SNPs did not show evidence of substantive over dispersion ($\lambda_{GC}$ values 1.02–1.08; Supplementary Fig. 2). Given the biological heterogeneity of ALL, as evidenced by subtype-specific associations at a number of previously published regions[9,12,18], we analysed the association between genotype and all B-ALL cases, and the common subtypes of HD and ETV6-RUNX1-positive ALL. Risk loci that were genome-wide significant only with a particular ALL subtype were defined as subtype-specific associations.

Meta-analysis identified 16 risk loci above genome-wide significance ($P < 5 \times 10^{-8}$, by inverse-variance method based on a fixed-effects model), of which 10 are previously reported B-ALL risk loci (Fig. 1 and Supplementary Table 1). Of the six new genome-wide significant candidate risk loci, one was generic to all B-ALL, three were specific for high-HD ALL and two were

specific for ETV6-RUNX1-positive ALL (Supplementary Table 1). These six SNP associations were interrogated in an independent series of 2237 cases and 3461 controls (COG_SJ GWAS non-European American (EA); Supplementary Tables 2 and 3). Four of the six SNPs were validated in the replication series ($P < 0.05$, by additive logistic regression test): for all B-ALL at 9q21.3 (rs76925697, nearest gene TLE1), for HD ALL at 5q31.1 (rs886285, C5orf56) and 6p21.31 (rs210143, BAK1), and for ETV6-RUNX1-positive ALL at 17q21.32 (rs10853104, IGF2BP1) (Table 1 and Supplementary Tables 2, 4 and 5). In addition to providing further evidence for the 21q22.2 association for all B-ALL[14], we also identified a subtype-specific association for HD with rs9976326 (Table 1 and Supplementary Tables 1, 4 and 5).

Next, we performed a conditional analysis on the sentinel risk SNP at each locus to search for further independent signals at new and previously reported risk regions. We confirmed the presence of previously reported dual association signals at 9p21.3 (CDKN2A/B) and 10p12.2 (PIP4K2A) (Supplementary Table 6). In addition, independent risk variants were identified at 21q22.2 (ERG) and 7p12.2 (IKZF1) (Supplementary Table 7 and Supplementary Figs. 3, 4, 5, 6 and 7).

### Functional annotation of new risk loci.
To gain insight into the biological basis of association signals, we examined the epigenetic landscape of risk regions in B cells. For each of the new risk regions, we evaluated chromatin profiles using ChromHMM, ATAC-seq data in primary B cells from the Roadmap Epigenomics consortia[23], and the GM12878 lymphoblastoid cell line from ENCODE[24,25] (Fig. 2 and Supplementary Figs. 3, 4, 5, 6, 7). As the strongest associated GWAS SNP may not represent the causal variant, we examined variants in linkage disequilibrium (LD) with the top risk SNP in each region (defined by $r^2 > 0.8$, $P < P_{min} \times 50$; Supplementary Table 8). Genomic spatial proximity and chromatin looping between non-coding DNA and target genes are key to gene regulation; we therefore interrogated promoter capture Hi-C (CHiC) data from naive B cells[26] (Supplementary Table 9) as well as Hi-C and H3K27Ac ChIP data in human ALL cells[27] (Supplementary Fig. 8). We also sought to identify target genes by performing quantitative trait locus (QTL) analysis of mRNA expression (eQTL) data from GTEx[28], Blood eQTL[29], MuTHER[30] and CAGE[31] databases, and DNA methylation (mQTL) (Supplementary Table 10). We annotated risk loci with variants mapping to haematopoietic transcription factor (TF)-binding sites (Fig. 2, Supplementary Figs. 3, 4, 5, 6 and 7, and Supplementary Table 11). Using Summary data-based Mendelian Randomisation (SMR) analysis, we examined for pleiotropy between GWAS signal and cis-eQTL for genes within 1 Mb of the sentinel SNP to identify a possible causal relationship between gene expression and disease (Supplementary Tables 12 and 13).

Lead SNPs at 6p21 are located within an intron 1 kb downstream of the BAK1 transcription start site and possess histone marks characteristic of active promoter activity and open chromatin accessibility (Fig. 2a). The top SNP, rs210143, falls within a TF-binding cluster and the C-risk allele is associated with reduced BAK1 expression ($P_{Blood} = 3.3 \times 10^{-310}$, by linear regression test). SMR analysis confirmed a significant association with BAK1 expression and ALL consistent with a likely causal relationship (Supplementary Table 12). The 6p21 association was confined to HD ALL only, whereas risk variants did not reach genome-wide significance for either ETV6-RUNX1 or all B-ALL. BAK1 was not differentially expressed in leukaemic blasts from any ALL subtype (Supplementary Fig. 9).

The HD ALL-specific association at 5q31 (C5orf56) localises to genomic regions featuring ChIP-seq marks indicative of

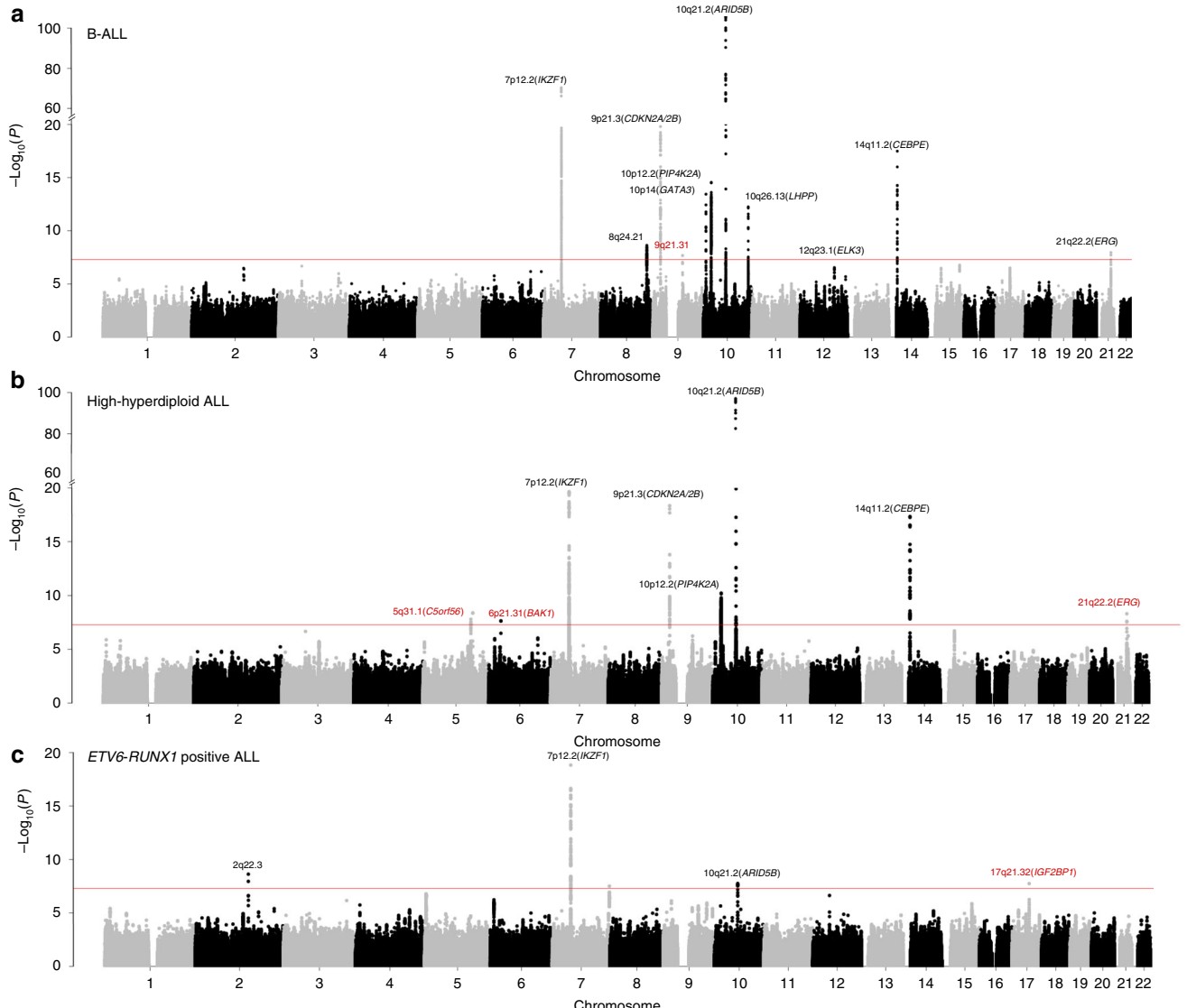

**Fig. 1** Manhattan plots of association for **a** B-ALL, **b** high-hyperdiploid ALL and **c** *ETV6-RUNX1*-positive ALL. *y*-axis shows genome-wide *P*-values (two-sided, calculated using SNPTEST v2.5.2 assuming an additive model) of > 6 million successfully imputed autosomal SNPs in 5321 cases and 16,666 controls. The *x*-axis shows the chromosome number. The red horizontal line represents the genome-wide significance threshold of $P = 5.0 \times 10^{-8}$. New associations are labelled in red. Other risk loci were reported in previous GWAS using subsets of ALL cohorts included herein.

regulatory elements. Although SNP rs2522044 is eQTL for *SLC22A4* and *C5orf56* ($P_{Blood} = 8.8 \times 10^{-51}$ and $7.3 \times 10^{-12}$, respectively, by linear regression test), a looping interaction between the top SNP rs886285 and the immune regulatory gene *IRF1* was observed (Fig. 2b and Supplementary Table 9). SMR analysis did not reveal any association with *C5ORF56*, *SLC22A4* or *IRF1* expression, nor did these genes show subtype-specific expression in ALL blasts (Supplementary Table 12).

Risk SNPs at 17q21 localising to the second intron of *IGF2BP1* lack evidence of cis-regulatory activity. However, the strongest associated SNP, rs10853104, maps to a TF-binding cluster and is predicted to disrupt a conserved CTCF-binding motif (Fig. 2c), suggesting an influence on topological-associated domain structure. As the 17q21 association was unique to *ETV6-RUNX1*-positive ALL, we investigated the relationship between ALL subtype and expression of genes within 1 Mb of rs10853104. *ETV6-RUNX1*-positive ALL cells showed significant overexpression of *IGF2BP1* compared with other ALL subtypes

(Supplementary Figs. 10 and 11; $P = 3.68 \times 10^{-23}$, by two-sided Wilcoxon's rank-sum test).

The lead SNP rs76925697 at a new B-ALL risk locus in 9q21 resides 500 kb centromeric to *TLE1* within a genomic region devoid of chromatin marks indicative of regulatory function (Fig. 2d). We also did not observe any evidence for eQTLs or TF binding. However, in ALL cells, the region containing the risk SNP showed strong looping with a distal enhancer within *TLE1* (Supplementary Fig. 8). Finally, we identified an HD ALL-specific association at the previously reported 21q22 locus within intron 3 of *ERG*. Notably, the T-risk allele of the lead SNP rs9976326 is predicted to disrupt binding of the haematological TF AML1/RUNX1 and is associated with reduced gene methylation.

Transcriptome-wide association studies (TWASs) investigating the association of genetically predicted gene expression with disease can identify new susceptibility genes by aggregating evidence across variants, thereby increasing study power. We

**Table 1 Summary of results for genome-wide significant childhood ALL risk loci.**

| CHR | SNP (Subtype) | Locus (gene) | Position (BP) | Risk allele | RAF | OR (95% CI) | *P*-value |
|---|---|---|---|---|---|---|---|
| 2 | rs17481869(*ETV6-RUNX1*) | 2q22.3 | 146124454 | A | 0.08 | 1.74 (1.45–2.09) | $2.37 \times 10^{-09}$ |
| 5 | *rs886285 (High-Hyperdiploidy) | 5q31.1 (*C5orf56*) | 131765206 | T | 0.34 | 1.29 (1.18–1.41) | $1.56 \times 10^{-08}$ |
| 6 | *rs210143 (High-Hyperdiploidy) | 6p21.31 (*BAK1*) | 33546930 | C | 0.73 | 1.30 (1.19–1.43) | $2.21 \times 10^{-08}$ |
| 7 | rs17133805 | 7p12.2 (*IKZF1*) | 50477514 | G | 0.32 | 1.65 (1.56–1.74) | $5.28 \times 10^{-71}$ |
| 8 | rs75777619 | 8q24.21 | 130185176 | G | 0.12 | 1.26 (1.17–1.36) | $2.30 \times 10^{-09}$ |
| 9 | *rs76925697 | 9q21.31 | 83747371 | A | 0.96 | 1.52 (1.31–1.76) | $2.11 \times 10^{-08}$ |
| 9 | rs113650570 | 9p21.3 (*CDKN2A*) | 21976402 | A | 0.02 | 2.32 (2.03–2.65) | $8.06 \times 10^{-35}$ |
| 10 | rs10821936 | 10q21.2 (*ARID5B*) | 63723577 | C | 0.33 | 1.80 (1.71–1.89) | $1.19 \times 10^{-106}$ |
| 10 | rs3824662 | 10p14 (*GATA3*) | 8104208 | A | 0.19 | 1.29 (1.21–1.38) | $3.57 \times 10^{-14}$ |
| 10 | rs2296624 | 10p12.2 (*PIP4K2A*) | 22856946 | C | 0.67 | 1.25 (1.18–1.32) | $2.79 \times 10^{-15}$ |
| 10 | rs12779301 | 10q26.13 (*LHPP*) | 126292655 | C | 0.66 | 1.22 (1.15–1.29) | $5.72 \times 10^{-13}$ |
| 12 | rs4762284 | 12q23.1 (*ELK3*) | 96612762 | T | 0.32 | 1.15 (1.12–1.19) | $3.75 \times 10^{-07}$ |
| 14 | rs2239630 | 14q11.2 (*CEBPE*) | 23589349 | A | 0.45 | 1.28 (1.22–1.35) | $1.72 \times 10^{-21}$ |
| 17 | *rs10853104 (*ETV6-RUNX1*) | 17q21.32 (*IGF2BP1*) | 47092076 | T | 0.47 | 1.33 (1.21–1.47) | $1.82 \times 10^{-08}$ |
| 21 | rs9976326 (High-Hyperdiploidy) | 21q22.2 (*ERG*) | 39776485 | T | 0.25 | 1.33 (1.21–1.46) | $4.79 \times 10^{-09}$ |

*BP* base pair, *CHR* chromosome, *CI* confidence intervals, *OR* odds ratio, *RAF* risk allele frequency. OR and CI are derived from current meta-analysis. *New loci discovered in current meta-analyses. Other risk loci were reported in previous GWAS using subsets of ALL cohorts included herein.

performed a TWAS integrating genomic and expression data[32]. This analysis confirmed the risk loci described above but did not identify any additional associations independent of GWAS signals, which were statistically significant (Supplementary Figs. 12 and 13).

To implicate recurrent disruption of TF-binding sites at ALL risk loci genome wide, we performed TF-binding enrichment analysis as *per* Cowper-Sal-lari et al.[33]. This analysis identified over-representation of TF binding at risk SNPs compared with a random SNPs subset. A number of TFs somatically mutated in B-ALL, including *PBX1* (Benjamini–Hochberg corrected *P*-value [$P_{BH}$] = 0.007), *TCF3* ($P_{BH}$ = 0.007), *ETS1* ($P_{BH}$ = 0.009), *RUNX1* ($P_{BH}$ = 0.012) and *ERG* ($P_{BH}$ = 0.030) (Supplementary Fig. 14) were enriched at risk loci providing evidence that germline variation and somatic alterations may impact on the same pathways. In addition, we identified BRD4 ($P_{BH}$ = 0.007) and NR3C1 ($P_{BH}$ = 0.009) binding sites as significantly enriched at risk loci, suggesting their disruption contributes to leukaemogenesis.

**Relationship between new risk alleles and clinical features.** We did not find an association between sex or age at diagnosis of ALL with the new risk SNPs using case-only analysis. We also found no statistically significant relationship between SNP genotype and patient outcome using data from German and COG_SJ GWAS cohorts[20,34]. A failure to demonstrate additional relationships may, however, be reflective of limited statistical power.

**Contribution of risk SNPs to heritability.** Using LD-adjusted kinships (LDAK)[35], the heritability of ALL ascribable to all common variation was identified as 21% (SD ± 0.065) (Supplementary Table 14). Together, the risk loci identified so far accounted for 31% of the total variance in genetic risk of ALL (Supplementary Table 15). To assess the collective impact of all identified risk SNPs we constructed polygenic risk scores (PRS) considering the combined effect of all risk SNPs modelled under a log-normal relative risk distribution after correcting the *Z*-scores for Winner's curse using FIQT[36]. Based on their PRS score, an individual in the top 1% of genetic risk would have a 4.7-fold increased risk of ALL when compared with an individual with median genetic risk (Supplementary Fig. 15).

**Discussion**
Our analysis provides evidence of four new associations with the risk of developing ALL. Besides providing additional evidence for

genetic susceptibility to ALL, these new risk loci provide further insights into the biological basis of ALL development. Integrating information from Hi-C data with chromatin profiling and eQTL/mQTL data implicates a number of genes with strong a priori evidence as the functional basis of associations, e.g., at 6p21.31 the pro-apoptotic protein *BAK1*, at 21q22.2 the haematological ETS TF *ERG* and at 17q21.32 proliferation factor *IGF2BP1*. Conditional analysis revealed two novel secondary associations at 7p12.2 (*IKZF1*) and 21q22 (*ERG*), in addition to the previously identified signals at 10p12.2 (*PIP4K2A*) and 9p21.3 (*CDKN2A/2B*). Two of the genome-wide significant associations from our discovery meta-analysis were not replicated. This may be the consequence of a different population allelic structure between cohorts of different ancestry (Europeans in the discovery and non-Europeans in the replication) or population-specific associations[14]. Our recently discovered T-ALL risk locus *USP7* was also not significant in this GWAS because of its lineage-specific effect on ALL susceptibility[15].

*BAK1* is essential for B-cell homoeostasis and its knockout mice accumulate immature and mature follicular B cells. *BAK1* induces apoptosis by binding to and antagonising anti-apoptotic proteins, including *BCL2*[37–39]. Reduced *BAK1* expression relieves repression of *BCL2*, inhibiting apoptosis and conferring a pro-survival advantage[40]. Proximity of the lead 6p21 risk variant, rs210143-T, to the *BAK1* promoter and a strong negative association with expression suggests decreased *BAK1* promotes ALL leukaemogenesis. Notably, the 6p21 region is also pleiotropic, influencing chronic lymphocytic leukaemia (CLL) and testicular cancer risk. Moreover, the strongest association for both CLL and ALL is rs210143, suggesting a similar mechanistic basis.

The 12q21 association at *IGF2BP1* is specific for *ETV6-RUNX1*-positive ALL and this subtype also significantly over-expresses *IGF2BP1*[41]. We did not observe a significant association between *IGF2BP1* genotype and its expression in *ETV6-RUNX1* ALL, plausibly because the subtle effects of this germline risk variant on *IGF2BP1* transcription were masked by the drastic upregulation as a result of *ETV6-RUNX1* fusion. The subtype-specific nature of the association may be explained by the observation that in *ETV6-RUNX1* positive ALL *IGF2BP1* binds to the *ETV6-RUNX1* transcript increasing its stability and expression[42]. *IGF2BP1* has been implicated in promoting proliferation and cell survival via the post-transcriptional regulation of a number of genes including *KRAS*, *MYC* and *PTEN*[43].

Our analysis confirms the ALL association at 21q22 (*ERG*) recently reported in Hispanics[44]. In addition, we report a new

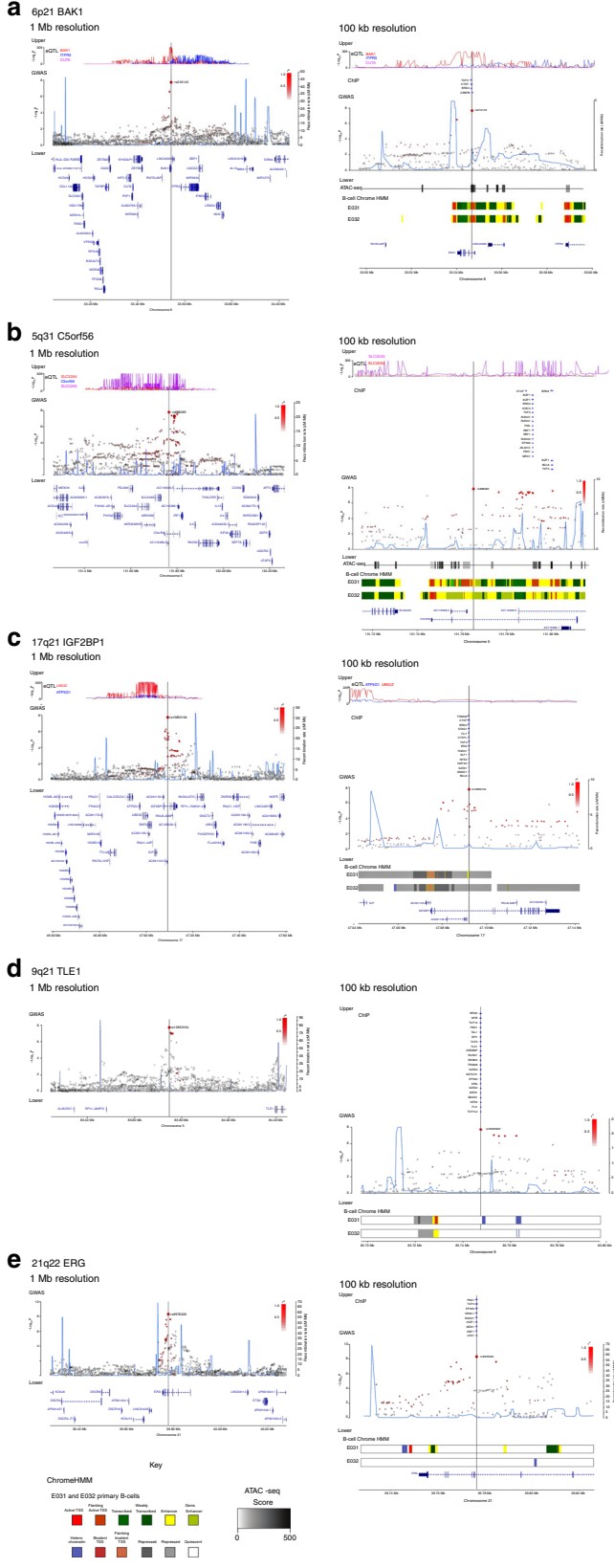

**Fig. 2** Regional plots of association results and recombination rates for the newly identified risk loci. **a** 6p21 (rs210143), **b** 5q31 (rs886285), **c** 17q21, **d** 9q21.3 (rs76925697), **e** 21q22 (rs9976326). Loci are shown at both 1 Mb (left) and 100 kb (right) resolutions. Upper panes show FDR corrected eQTL P-values extracted from the Blood database; ChIP transcription factor binding sites shown as blue bars. GWAS pane shows plots show association $-\log_{10}P$-values (left y-axis) of SNPs shown according to their chromosomal positions (x-axis). Light blue line shows recombination rates in (cM/Mb) from UK10K Genomes Project (right y-axis). Lead SNPs are denoted by large circles labelled by rsID. Colour intensity of each symbol reflects LD, white ($r^2 = 0$), dark red ($r^2 = 1.0$). Genome coordinates are from NCBI human genome GRCh37. Lower pane shows chromatin-state segmentation tracks (ChromHMM) from primary B cells and gene positions from Gencode v27 comprehensive gene annotation. Where no significant results were obtained upper and lower panes are omitted.

domain-containing TF important for normal hematopoietic development. Somatic alteration of *ERG* is recurrent in ALL and rs9976326 is in close proximity to hotspot deletions[45].

Although risk SNPs at 5q31 reside in *C5orf56*, which has no established role in B-cell biology, Hi-C interactions implicate the TF IRF1, which is required for normal T-cell development and is deleted in 50% of acute myelogenous leukaemia[46,47]. The intergenic region at 9q21.3 (near *TLE1*) has no clear candidate and the biological basis of the association is unclear.

TF-enrichment analysis revealed gene *BRD4* with no previous indication from germline or somatic studies in ALL and the gene *NR3C1* whose alterations are associated with poor outcome and high risk in ALL patients[48]. BRD4, a member of the BET protein family, is a transcriptional co-activator that binds acetylated histones recruiting TFs to DNA. BRD4 has been found to co-localise with the lymphoid TFs SPI1, FLI1, ERG, MYB and CEBPα/β[49], and this may account for its enrichment at risk loci. Several groups have shown activity of BET inhibitors in AML cells lines[50,51]. NR3C1 is the glucocorticoid receptor, the target of the immunomodulatory hormones glucocorticoids, including cortisol, and drugs including dexamethasone and prednisone. The effect of these compounds is potent immune-suppression and reduced inflammation. Glucocorticoid treatment reduces circulating B-cell numbers[52] and induces cell death in ALL cells by lowering the expression of B-cell survival factors[53]. Further validation will be required to establish a role of disrupted NR3C1 signalling in the genesis of ALL.

Deciphering the functional consequences of risk loci is inherently challenging, as analyses are complicated by background haplotype structure. We have relied in part on integration of GWAS signals with in silico and publicly accessible epigenetic data; hence, these predictions require experimental verification through functional assays in the future.

In summary, our study provides further evidence for inherited susceptibility to ALL and support for subtype specificity at risk loci. The different subtypes of B-ALL presumably reflect the different aetiology and evolutionary trajectories of progenitor cells influenced by inherited variation. Our findings further support a model of ALL susceptibility based on transcriptional dysregulation consistent with altered B-cell differentiation, where dysregulation of apoptosis and cell cycle signalling features as recurrently modulated pathways. Genes elucidated from GWAS functional annotation may represent promising therapeutic targets for drug discovery. Finally, although our GWAS meta-analysis is the largest of its kind, greater sample sizes are likely to uncover additional associations underscoring the need for collaborative analyses.

HD ALL association with rs9976326. The SNP rs9976326 and the top SNP reported in Hispanics (rs2836371) are separated by 3 kb and correlated (pairwise LD values $r^2 = 0.52$, $D' = 0.85$ and $r^2 = 0.60$, $D' = 0.87$ in European and admixed Americans 1000 genomes populations, respectively). *ERG* encodes an *ETS*

## Methods

**Ethics.** Collection of samples and clinical information was undertaken with informed consent and ethical review board approval. Specifically, Medical Research Council UKALL97/99 trial by UK therapy centres and approval for UKALL2003 from the Scottish Multi-Centre Research Ethics Committee (REC:02/10/052), the UK Bloodwise Childhood Leukaemia Cell Bank, the United Kingdom Childhood Cancer Study, and University of Heidelberg; AALL0232 (clinicaltrials.gov NCT00075725)[54] and P9904/P9905/P9906 (NCT00005585/NCT00005596/NCT00005603)[55] from the Children's Oncology Group (COG); and Total Therapy XIIIB/XV (NCI-T93-0101D/NCT00137111)[56,57] from the St. Jude Children's Research Hospital. The diagnosis of ALL was established in accordance with World Health Organization guidelines.

**GWAS data.** The four GWAS datasets have been the subject of previous publications: (i) UK GWAS I—824 cases, 2699 controls from the 1958 British Birth Cohort and 2501 controls from the UK Blood Service controls[7]; (ii) German GWAS—1155 Berlin–Frankfurt–Münster (BFM) trial (1993–2004) cases, 2132 Heinz Nixdorf Recall study controls[9]; (iii) UK GWAS II—1021 cases from Medical Research Council UK ALL-2003 and ALL-97/99 trials, 2976 PRACTICAL Consortium and 4446 Breast Cancer Association Consortium controls[12]; (iv) COG_SJ GWAS—2,879 cases of European ancestry from the COG AALL0232, COG P9904/P9905/P9906, St. Jude Total Therapy XIIIB/XV and 2057 non-ALL controls of European ancestry from the Multi-Ethnic Study of Atherosclerosis (MESA) study (dbGAP phs000209.v9)[13,18,20].

The replication study included 2237 cases and 3461 non-ALL controls of non-European ancestry from the same cohort as COG_SJ GWAS.

The UK GWAS I, UK GWAS II and German GWAS series were genotyped using Illumina Human 317K Human OmniExpress-12v1.0 or Infinium OncoArray-500K arrays. The COG_SJ GWAS and replication series were genotyped using Affymetrix Human SNP 6.0 (St. Jude Total XVI, COG P9904/9905, MESA) and Affymetrix GeneChip Human 500k Mapping arrays (St. Jude Total XIIIB/XV and COG P9906).

**Statistical analysis of GWAS data.** Analyses were undertaken using R v3.2.3[58], PLINK v1.9[59], SNPTEST v2.5.2[22] and IMPUTE v2.3[60] software. Standard quality-control measures were applied to each GWAS[61]. Specifically, individuals with low call rate ($< 95\%$) as well as all individuals with non-European ancestry (using the HapMap version 2 CEU, JPT/CHB and YRI populations (and Native American in COG_SJ dataset) as a reference) were excluded for discovery GWAS and meta-analysis. SNPs with call rate $< 95\%$ were excluded or showed deviation from Hardy–Weinberg equilibrium ($P < 10^{-5}$). Appropriateness case–control matching was evaluated using Q–Q plots inflation test statistics. The inflation factor $\lambda$ was calculated to indicate the degree of genomic inflation, by dividing the median of the test statistics by the median expected values from a $\chi^2$ distribution with 1 degree of freedom (Supplementary Fig. 2). Prediction of the untyped genotypes was carried out using 1000 Genomes Project (Phase 1) and UK10K as reference[62,63]. To account for genomic inflation post imputation, top Eigenvectors from the principal component analysis were used as covariates in the final association analysis[64]: the top two and five Eigenvectors for the UK_German GWAS and the COG_SJ GWAS, respectively. No further adjustments for P-values were applied. The association between each SNP and risk was calculated assuming an additive model and meta-analyses were performed using META v1.7[21,22]. Association meta-analyses only included SNPs with info score $> 0.8$, imputed call rates $> 0.9$ and MAFs $> 0.01$. We calculated Cochran's Q statistic to test for heterogeneity and the $I^2$ statistic to quantify the proportion of the total variation that was caused by heterogeneity.

In COG_SJ dataset, genetic ancestry (European [CEU], African [YRI], East Asian [JPT/CHB] and Native American) was determined by using ADMIXTURE (version 1.3.0)[65] with the sum of these four ancestries being 100% for any given subject. EA, African American and Asian were defined as having $> 95\%$ European genetic ancestry, $> 70\%$ African ancestry and $> 90\%$ Asian ancestry, respectively. Hispanics were individuals for whom Native American ancestry was $> 10\%$ and greater than African ancestry, as previously described[18]. Using a large reference panel of human haplotypes from the Haplotype Reference Consortium (HRC r1.1 2016)[66] in Michigan Imputation Server[66,67] with ShapeIT (v2.r790)[68] as the phasing tool, we imputed untyped SNPs genome-wide. SNPs were excluded if (1) imputation quality metric $R^2 < 0.3$ (indicating inadequate accuracy of the imputed genotype); (2) minor allele frequency in cases and controls $< 0.01$; (3) HWE $P < 1 \times 10^{-5}$ in cases and controls classified as European American. Using Q–Q plots inflation test statistics, we estimated an inflation factor $\lambda$ of 1.09 in the replication series.

The discovery GWAS P-value was thresholded at $5 \times 10^{-8}$ for genome-wide significance and replication P-value was thresholded at 0.05 for validation. For all four variants validated in the replication analysis, we estimated a false discovery rate $< 5\%$ with nominal P-value $< 0.05$, using Benjamini–Hochberg procedure. We performed the same statistical analyses for all the datasets unless specifically stated.

**Summary Mendelian randomisation analysis.** SMR analysis was conducted as per Zhu et al.[69] The most significant eQTL or mQTL for each gene was used as an instrumental variable to test for an association between expression levels of the gene and B-ALL using summary statistics from the meta-analysis GWAs dataset. The expression levels of the gene identified should be significantly associated with the disease as a result of true pleiotropy as opposed to correlation due to linkage between the GWAS variants and functional eQTL variants; accordingly, the heterogeneity in dependent instruments (HEIDI) analysis was performed as per Zhu et al.[69] Publicly available eQTL data were extracted from the CAGE eQTL dataset (peripheral blood, $n = 2765$)[31], GTEx eQTL v7, whole blood ($n = 369$) and Epstein-Barr Virus-transformed lymphocytes ($n = 117$), and blood eQTL datasets[29,70,71]. To investigate regulatory elements associated with B-ALL, we utilised the methylation QTL datasets Aberdeen (Blood, $n = 639$) and UCL (Blood, $n = 665$)[72]. All eQTL or mQTL summary datasets were pruned to only those probes with $P_{eQTL/mQTL} < 5 \times 10^{-8}$. GWAS summary statistics files were generated from the meta-analysis of UK GWAS I, UK GWAS II, German GWAS and COG_SJ datasets. Reference files were generated by merging 1000 genomes phase 3 and UK10K (ALSPAC and TwinsUK) data. Summary eQTL files for the GTEx samples were generated from downloaded v7p 'all_SNPgene_pairs' files. Only probes with eQTL $P < 5.0 \times 10^{-8}$ were considered in the SMR analysis. HEIDI test P-values $< 0.05$ were taken to indicate significant heterogeneity.

**Association test of predicted gene expression with ALL risk.** Associations between predicted gene expression and ALL risk were examined using MetaXcan, accounting for LD[32]. SNP weights and their respective covariance for all GTEx tissues were obtained from predict.db (http://predictdb.org/), which is based on GTEx version 7 eQTL data. To combine S-PrediXcan data across the different tissues taking into account tissue–tissue correlations, we used S-MultiXcan. To determine whether associations between genetically predicted gene expression and ALL risk were influenced by variants previously identified by GWAS, we performed conditional analyses adjusting for GWAS risk SNPs (Supplementary Table 16) predicted by GCTA-COJO stepwise logistic regression analysis[73,74]. Adjusted output files were provided as the input GWAS summary statistics for S-PrediXcan analyses as above.

**Functional-epigenetic annotation.** Promoter CHiC, chromatin-state annotation and TF analyses were performed on lead SNPs, defined as any SNP with a P-value $< P(min) \times 50$ and $R^2 > 0.8$ from the lead SNP at a locus.

**eQTL data.** SNP gene expression associations were extracted from the Blood[29], CAGE[31], and MuTHER[70] eQTL datasets. Only associations from the Blood dataset are shown in Fig. 2.

**ATAC-seq.** Chromatin accessibility in the lymphoblastoid cell line GM12878 was extracted from GSE47753[75].

**Chromatin-state annotation.** Chromatin-state segregation data, analysed by ChromHMM, ae shown for the primary B-cell lines E031 and E032, and the lymphoblastoid cell line GM12878 from the roadmap[76] and Encode[24] projects, respectively.

**Promoter capture Hi-C.** Promoter-looping interactions were downloaded and filtered for a $-\log(\text{weighted } P) \geq 5$ in naive B cells only[26]. Interactions were called using CHiCAGO[77]. Interactions overlapping lead SNPs in each locus are reported.

**Hi-C and histone mark ChIP-seq in ALL cells.** Hi-C and H3K27Ac ChIP-seq were performed in human ALL cell line Nalm6 at St. Jude[27]. For Hi-C, the Nalm6 cell line was cultured under recommended conditions to about 80% confluence. Five million cells were crosslinked with 1% formaldehyde for 10 min at room temperature, then digested with 125 units of MboI and labelled with biotinylated nucleotides and were proximity ligated. After reverse crosslinking, ligated DNA was purified and sheared to 300–500 bp, then ligation junctions were pulled down with streptavidin beads and prepared as a general Illumina library. The Hi-C sample was sequenced paired-end 76 cycles on Illumina Hiseq 4000. For the H3K27ac ChIP-seq, a frozen cell pellet containing 10 million cells was sent to Active Motif for ChIP and library preparation. The sample was divided into an aliquot for ChIP using an antibody to H3K27ac (Active Motif) and an input control. Single-end sequencing was performed using an Illumina NextSeq 500 generating 76 cycles for each sequencing read. Histone acetylation mark and chromatin looping signals were directly downloaded from the NCBI GEO GSE115494 dataset. Loop interactions were called using HiCCUPS[78] from Juicer tools v1.12.01 under default parameters at a resolution of 5 kb and 10 kb. Enriched interaction was reported with a false discovery rate $< 0.1$.

**TF-enrichment analysis.** TF-binding enrichment analysis was performed according to the method of Cowper-Sal lari et al.[33] examining SNPs in LD with the sentinel SNP (i.e., $r^2 > 0.8$ and $D' > 0.8$). Publically available TF ChIP-seq data were obtained from ChIP-Atlas (http://chip-atlas.org/). TF-binding sites were filtered for those with a MACS peak Q-value $> 100$ and from cells lines with a 'blood'

annotation. Overlapping binding sites from the same ChIP target were merged. For each mark, the overlap of the SNPs and the binding sites was assessed to generate a mapping tally. A null distribution was produced by performing 10,000 permutations, randomly selecting LD blocks with the same number of SNPs as the test set, and calculating the null mapping tally. P-values were calculated by normalising the tallies to the median of the null distribution.

**Heritability analysis**. We used LDAK version 4.9[35] to estimate the polygenic variance (i.e., heritability) ascribable to all genotyped and imputed GWAS SNPs. Heritability ascribed to all the genotyped and imputed SNPs was calculated from summary data after filtering; information score filtering ($>0.99$), allele frequency ($>0.01$) and Hardy–Weinberg deviation ($P < 1 \times 10^{-5}$), resulting in 1,553,634 SNPs for analyses. SNP-specific weightings were calculated reflecting correlations across SNPs (predictors) using UK10K and 1000 genomes data, after adjusting for LD, MAF and genotype certainty.

**Contribution of genetic variance to familial risk**. Estimation of risk variance associated with each SNP was performed as per Pharoah et al.[79] For an allele ($i$) of frequency $p$, relative risk $R$ and log risk $r$, the risk distribution variance ($V_i$) is:

$$V_i = (1-p)^2 E^2 + 2p(1-p)(r-E)^2 + p^2(2r-E)^2 \quad (1)$$

where $E$ is the expected value of $r$ given by:

$$E = 2p(1-p)r + 2p^2 r \quad (2)$$

For multiple risk alleles, the distribution of risk in the population tends towards the normal with variance:

$$V = \Sigma V_i \quad (3)$$

The percentage of total variance was calculated assuming a familial risk of childhood ALL of 3.2 (95% confidence interval (CI) 1.5–5.9) as per Kharazmi et al.[80] All genetic variance ($V$) associated with susceptibility alleles is given as $\sqrt{3.2}$[80]. The proportion of genetic risk attributable to a single allele is:

$$V_i V^{-1}$$

Eighteen risk loci were included in the calculation of the PRS for childhood ALL by selecting the top SNP from the current meta-analysis from each previously published loci in addition to the two risk loci discovered in this study. The 11 variants are thought to act independently, as previous studies have shown no interaction between risk loci[7,9–11]. PRS were generated as per Pharoah et al.[79] assuming a log-normal distribution $LN(\mu,\sigma^2)$ with mean $\mu$ and variance $\sigma^2$. The population $\mu$ was set to $\sigma^2/2$, in order that the overall mean PRS was 1.0.

**Relationship between SNP genotype and ALL clinical features**. The relationship between SNP genotype and survival was analysed in the German AIEOP-BFM series, which consisted of 834 patients within the AIEOP-BFM 2000 trial. Patients were treated with conventional chemotherapy (i.e., prednisone, vincristine, daunorubicin, l-asparaginase, cyclophosphamide, ifosfamide, cytarabine, 6-mercaptopurine, 6-thioguanine and methotrexate), a subset of those with high-risk ALL were treated with cranial irradiation and/or stem cell transplantation. Events, for event-free survival, were defined as resistance to therapy, relapse, secondary cancer or death. Kaplan–Meier methodology was used to estimate survival rates, with differences between groups tested using the log-rank method (two-sided P-values). Cumulative incidences of competing events were calculated using the methodology of Kalbfleisch and Prentice, and compared using Gray's test. Cox regression analysis was used to estimate hazard ratios and 95% CIs adjusting for clinically relevant covariates. Similar analyses of SNP genotype with treatment response and outcome measures were performed in the COG_SJ series as reported previously[20,34]. No significant association was observed for these novel risk SNPs (i.e., $P > 0.05$)[20,34].

**Reporting summary**. Further information on research design is available in the Nature Research Reporting Summary linked to this article.

## Data availability

UK controls were obtained from the Wellcome Trust Case Control Consortium 2 (http://www.wtccc.org.uk/; 50.7% male;[81] WTCCC2:EGAD00000000022 and EGAD00000000024). Imputation reference panels are available from 1000 G phase I (ftp://ftp.1000genomes.ebi.ac.uk/vol1/ftp/release/20110521/) and the UK10K ($n = 3781$; EGAS00001000090, EGAD00001000195 and EGAS00001000108; www.uk10k.org). The UK GWAS I, UK GWAS II and German GWAS data for ALL cases are available through the European Genome-Phenome Archive website (EGA, https://ega-archive.org, EGAS00001003937, EGAS00001002809 and EGAS00001003936, respectively). The SJ_COG GWAS data for ALL cases are deposited in the NIH dbGAP (https://www.ncbi.nlm.nih.gov/gap/) under phs000638.v1.p1 and phs000637.v1.p1. ATAC-seq dataset GSE47753_GM12878_ATACseq_50k_AllReps_ZINBA_pp08.bed.gz was downloaded from Gene Expression Omnibus (https://www.ncbi.nlm.nih.gov/geo/query/acc.cgi?acc=GSE47753). ChromHMM data for primary B-cell are available at http://egg2.wustl.edu/roadmap/data/byFileType/chromhmmSegmentations/ChmmModels/coreMarks/

jointModel/final/ and ChromHMM annotation for GM12878 is available from http://hgdownload.cse.ucsc.edu/goldenPath/hg19/encodeDCC/wgEncodeBroadHmm/wgEncodeBroadHmmGm12878HMM.bed.gz. Promoter CHiC data are available at https://osf.io/u8tzp/. eQTL and mQTL data for SMR analysis were downloaded from https://cnsgenomics.com. GTEx version 7 data are available at https://gtexportal.org/home/datasets. Requests for other data should be directed to the authors.

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

## Acknowledgements

In the UK, funding was provided by Bloodwise and Cancer Research UK (C1298/A8362). In the United States, this work was partly supported by National Institutes of Health Grant Numbers CA21765, CA98543, CA114766, CA98413, CA180886, CA180899, GM92666, GM115279, and GM097119, and the American Lebanese Syrian Associated Charities. We thank the patients and parents who participated in the Children's Oncology Group (COG) protocols included in this study, the clinicians and research staff at St. Jude Children's Research Hospital and COG institutions, Jeanette Pullen (University of Mississippi, Jackson, MS) for assistance in the classification of patients with ALL and Mark Shriver (Pennsylvania State University, University Park, PA) for sharing single-nucleotide polymorphism genotype data of the Native American references. M.Q. is supported by the Initial Funding for New PI of Fudan University, the National Natural Science Foundation of China (81973997) and the Program for Professor of Special Appointment (Eastern Scholar) at Shanghai Institutions of Higher Learning. S.P.H. is the Jeffrey E. Perelman Distinguished Chair in Pediatrics at The Children's Hospital of

Philadelphia. M.L.L. is the University of California, San Francisco Benioff Chair of Children's Health and the Deborah and Arthur Ablin Chair of Pediatric Molecular Oncology.

## Author contributions

J.J.Y. and R.S.H. designed the overall study. Association analysis and statistical data analysis were performed by J.V. and M.Q. Functional analysis was undertaken by J.B.S., J.V., W.Y. and M.Q. W.Y., B.K. and P.J.L. provided bioinformatics support. P.B. supervised the data production of UK GWAS II. J.A., A.V. and A.M. provided samples recruited on the ALL-97/99 and ALL-2003 trials. C.R.B., M. Stanulla, M. Schrappe and M.Z. provided samples through the Berlin–Frankfurt–Münster (BFM) trial (1993–2004); M. Stanulla and M.Z. conducted outcome analysis on BFM samples. E.A.R., C.-H.P., W.E.E., C.G.M., S.P.H., M.V.R. and M.L.L. supervised the sample collection and data production of the COG_SJ cohort. C.-H.P., W.E.E., A.Y., C.L., S.P.H., M.V.R., M.L.L., R.S.H. and J.J.Y. interpreted the data and the research findings. The manuscript was drafted by J.J.Y., R.S.H., J.V., M.Q. and J.B.S., and was reviewed by all of the co-authors.

## Competing interests

The authors declare no competing interests.
