## [Peer Review File · Nature Communications]

Reviewers' comments:

Reviewer #1 (Remarks to the Author):

In this manuscript, Vijayakrishnan et al. present the results of a meta-GWAS analysis of four previously studied cohorts of pediatric B-cell acute lymphoblastic leukemia (ALL) in children. By analyzing over 5300 samples (vs. >16600 controls), they confirm 10 of 11 previously identified risk variant for this disease and identify a novel variant at 9q21 as a new B-ALL risk locus. Analysis of B-ALL subclasses confirmed previously identified risk variants and revealed novel subtype-specific associations. Three new loci were associated with high-hyperdiploid ALL (5q31, 6p21 and the variant at 21q22 that was previously reported for B-ALL, but not for this disease subtype). Likewise, a novel variant at 17q21 near the IGF2BP1 gene was significantly associated with ETV6-RUNX1-positive B-ALL.

Newly discovered regions are functionally annotated with datasets on gene expression, chromatin accessibility and -interaction, transcription factor binding, eQTL, clinical parameters and others.

The new associations identified in this work will clearly contribute to a better genomic understanding of pediatric B-ALL and its subtypes.

The manuscript is well-written, the methodology is clearly explained and the results are presented in an intuitive fashion.

Yet, while the authors invested considerable effort in putting the newly identified variants into biological context by aggregating multiple additional datasets from various sources, the results still remain descriptive and correlative. Thus, I do not agree with the term "Identification of [...] regulatory mechanisms..." in the title of the manuscript.

The broad impact of this work would be greatly enhanced if the authors could provide a more detailed investigation about potential mechanistic aspects of at least one of the newly identified variants in a biologically relevant system.

Minor points:

- Can the authors discuss why the previously identified 16p13 (USP7) variant was not found in their analysis?

- Could the authors use colors instead of arrows to illustrate known vs. new variants identified in the present study in Figure 1?

The order of panels in Figure 2 should be re-assembled to follow the narrative in the text.

- It is difficult to distinguish the 21 colors of the Chrome HMM annotation throughout Figure 2. This could be simplified.

- The axis label in the right Y-axis in figure 2A (left panel) is missing.

Reviewer #2 (Remarks to the Author):

The manuscript from Vijayakrishnan et al. provides evidence for additional ALL risk loci and suggestions for regulatory mechanisms. The discovery phase for this study is a meta analyses of 4 previous European GWAS sets. The replication is based on analyses of an independent set of non-European ALL set. Overall the study provides potentially useful analyses that may facilitate further understanding the genetics of ALL. There are aspects of the study that should be addressed.

1. The replication dataset and data should be described better. This should include a) % of each ancestry in the cases and controls; b) risk allele frequency (and note whether risk allele is same as in discovery); and c) assessment of lambda if there is genome-wide SNP data (and PCA or other correction if warranted).

2. The replication statistics should adjust for multiple comparison e.g. Bonferroni correction would exclude the rs210143 (BAK1) from the "replicated" set. Much of the discussion and summary

centers on this result that is marginal.

3. Conversely, the non-replication of 2 or 3 of the loci that met the GWAS criterion may be due to the assessment in a different ancestry (e.g. even absence of the risk allele). This should be discussed.

4. It is not clear to this reviewer that all ten previously reported B-ALL risk loci are “confirmed” (page 4) rather than consistent with the previous reports. The question is whether the previous reported B-ALL risk loci were all identified in subjects independent of those used in the meta-analyses.

5. There is a substantial amount of speculative discussion concerning the integration of GWAS signals with potential epigenetic mechanisms. The limitations of these in silico compilations should be emphasized including what appears to be a complete lack of any data that indicate that variants on susceptibility haplotypes are associated with the epigenetics.

Reviewer #3 (Remarks to the Author):

Vijayakrishnan et al present a meta-analysis of 4 GWAS datasets to identify novel risk variants for B-ALL. The cohort includes an impressive number of cases making it the largest B-ALL GWAS reported. They identify 4 novel risk loci, one for B-ALL in general, 2 specific for hyperdiploid ALL and 1 specific for ETV6-RUNX1 ALL. They go on to examine publically available epigenetic data from experiments using primary B cells and a B-lymphoblastoid cell line in an attempt to provide support for the likely relevance of these loci to B-ALL.

The authors have extensive experience in the field and the methods employed in the GWAS meta-analysis seem robust, but would require review from a statistician with expertise in this area. The manuscript is concise and clearly written. I am less convinced by the analysis of the epigenetic data, which fails to provide conclusive evidence for the importance of these SNPs in relation to gene expression in a definitively causative fashion.

Comments

1. The authors published a very similar paper earlier this year (Vijayakrishnan et al, Nat Comm 2019) which analyzed three of the 4 included GWAS datasets (the 2 UK studies and the German study), included 2,442 cases and identified 2 novel risk loci. This paper extends this analysis through the inclusion of the COG dataset adding a further 2,879 cases. I think the exact datasets included and the link to the recent paper should be made clearer in the introduction. While the paper does identify additional significant variants, and confirm their previous findings, the novelty is somewhat tempered by the previous paper.

2. Figure 1. What do the asterisks refer to genes such as BAK1?

3. It is noted that 4 of the 6 SNP were validated in the replication series (pg 4) with $P < 0.05$ – is this a significant P value at genome-wide significance, which was previously defined as $< 5 \times 10^{-8}$?

4. Figure 2 and associated legend require significant changes. The legend needs far more information to explain the lower part of each plot, and it takes a lot of work on behalf of the reader to work out what the authors are showing. Please explain why each sub-figure is split into 1) and 2). ATACseq and eQTL is not mentioned in the legend, nor the Y-axis for the eQTL section – does this P value include a correction for multiple testing (see Huang et al NAR 2018)? The ChromHMM data is not adequately explained and the same colors represent different things in different samples. Please order figures in relation to text; ie. 2A) 6p21, 2B) 5q31 etc.

5. The second half of the results, analyzing publically-available epigenetic data provides rather tenuous support to the identified loci in some cases, and most of these data are circumstantial, descriptive and correlative. While rs210143 at the 6p21 locus does appear to fall within a region of active promoter activity, supporting a causal role, 5q31 lies in an area of largely closed chromatin, so what is its putative role in altering gene expression in this setting? Do the SNPs alter predicted

TF binding motifs?

6. Supp Figure 9 and 10 show that IGF2BP1 is overexpressed in ETV6-RUNX1 ALL. Do these cases carry the rs10853104 SNP and is it associated with increased IGF2BP1 expression?

7. The authors mention that there was no statistically significant relationship between SNP genotype and outcome in the German data. Why was this analysis not extended to the combined dataset as outcome data are available on all these cohorts?

8. Standard eQTL studies tend to link gene expression with SNP genotype, but do not necessarily take into account allelic expression. If the SNPs described truly alter gene expression, then one would expect allelic imbalance in gene expression (when SNP is heterozygous) - this is potentially detectable by analysis of coding heterozygous SNPs in the gene, or its 3'UTR. Similarly, TF binding would be allelically skewed. Have the authors looked into this?

9. Given this study offers only a minor incremental advance from their recent Nat Comms paper, this study would have massively benefited from functional studies, for instance altering the SNPs with CRISPR in cell lines and analyzing gene expression to show true causation.

August 14th, 2019

We appreciate the constructive review of our manuscript entitled, “Identification of new risk loci and regulatory mechanisms influencing genetic susceptibility to acute lymphoblastic leukaemia in children” (NCOMMS-19-12126), and the invitation to revise. We have carefully considered all of the reviewer comments, performed substantial additional analyses and experiments with new data included, to address each of the questions/critiques as outlined below (marked in red). We have thus modified the manuscript accordingly (highlighted in yellow). For ease of evaluation, we have described our responses directly within the template of the reviewers’ comments.

Reviewer #1 (Remarks to the Author):

In this manuscript, Vijayakrishnan et al. present the results of a meta-GWAS analysis of four previously studied cohorts of pediatric B-cell acute lymphoblastic leukemia (ALL) in children. By analyzing over 5300 samples (vs. >16600 controls), they confirm 10 of 11 previously identified risk variant for this disease and identify a novel variant at 9q21 as a new B-ALL risk locus. Analysis of B-ALL subclasses confirmed previously identified risk variants and revealed novel subtype-specific associations. Three new loci were associated with high-hyperdiploid ALL (5q31, 6p21 and the variant at 21q22 that was previously reported for B-ALL, but not for this disease subtype). Likewise, a novel variant at 17q21 near the IGF2BP1 gene was significantly associated with ETV6-RUNX1-positive B-ALL.

Newly discovered regions are functionally annotated with datasets on gene expression, chromatin accessibility and -interaction, transcription factor binding, eQTL, clinical parameters and others.

The new associations identified in this work will clearly contribute to a better genomic understanding of pediatric B-ALL and its subtypes.

The manuscript is well-written, the methodology is clearly explained and the results are presented in an intuitive fashion.

1. Yet, while the authors invested considerable effort in putting the newly identified variants into biological context by aggregating multiple additional datasets from various sources, the results still remain descriptive and correlative. Thus, I do not agree with the term “Identification of [...] regulatory mechanisms...” in the title of the manuscript.

As suggested, we have now changed the title to “Identification of four novel associations for acute lymphoblastic leukaemia risk”.

2. The broad impact of this work would be greatly enhanced if the authors could provide a more detailed investigation about potential mechanistic aspects of at least one of the newly identified variants in a biologically relevant system.

We agree with this reviewer. However, comprehensive mechanistic follow-up on the novel risk loci would be a very significant new effort and is beyond the scope of the current manuscript.

To address this concern and as suggested by this reviewer, we have analyzed in-house data at St. Jude for Hi-C and H3K27Ac ChIP in human ALL cell line (Nalm6). In contrast to naïve B cell data presented in the original manuscript, our new results provided a direct assessment of potential interactions of the risk variants with regulatory elements, in the most biologically relevant context. At 3 of the 4 novel risk loci (5q31.1, 6p21.31, and 17q21.32), we observed looping between risk variants and adjacent putative regulatory elements in ALL cells in a pattern similar to that in naïve B cells (**Figure 2**). Interestingly, the 9q21.31 locus is void of any looping signal in normal B cells, but in ALL cells there is a strong interaction of the risk variant with a downstream enhancer in the *TLE*

gene (477 Kb distal to the GWAS hit). We have now included these new results in the Text (**Pages 5 and 6**) and as **Supplemental Figure 8**.

Minor points:

3. Can the authors discuss why the previously identified 16p13 (USP7) variant was not found in their analysis?

This risk locus was identified from analysis of T-cell ALL (*J. Natl. Cancer Inst* 2019 111(12) djz043) while our current analysis is exclusively restricted to B-cell ALL. In that study, we also showed that the *USP7* risk variant was highly lineage specific with no association with B-ALL. Hence we would not expect to see any signal at the *USP7* locus in the current GWAS. This is now discussed on **Page 8**.

4. Could the authors use colors instead of arrows to illustrate known vs. new variants identified in the present study in Figure 1?

As requested, we revised **Figure 1** to highlight all new loci in red with known risk genes in black.

5. The order of panels in Figure 2 should be re-assembled to follow the narrative in the text.

As requested, panels in **Figure 2** have been rearranged to follow the narrative.

6. It is difficult to distinguish the 21 colors of the Chrome HMM annotation throughout Figure 2. This could be simplified.

We originally adhered to the colour scheme defined by the ChromHMM software. As suggested, we have now simplified this by merging functional groupings.

7. The axis label in the right Y-axis in figure 2A (left panel) is missing.

We apologize for the omission and axis label is now provided.

Reviewer #2 (Remarks to the Author):

The manuscript from Vijayakrishnan et al. provides evidence for additional ALL risk loci and suggestions for regulatory mechanisms. The discovery phase for this study is a meta analyses of 4 previous European GWAS sets. The replication is based on analyses of an independent set of non-European ALL set. Overall the study provides potentially useful analyses that may facilitate further understanding the genetics of ALL. There are aspects of the study that should be addressed.

1. The replication dataset and data should be described better. This should include a) % of each ancestry in the cases and controls; b) risk allele frequency (and note whether risk allele is same as in discovery); and c) assessment of lambda if there is genome-wide SNP data (and PCA or other correction if warranted).

As requested we now provide further information of the replication series, as follows:

- We added a new **Supplementary Table 3** to list the number of subjects in cases and controls of each ancestry;
- We added risk allele frequency (in cases and matching controls based on top 10 eigenfactors) in the replication cohort in **Supplementary Table 2**. For all 4 validated loci the direction of association was the same in the replication cohort as in the discovery cohort.
- As suggested, we examined potential population stratification in the replication cohort, with an estimated lambda of 1.09 (**Page 12**). This is comparable to findings from our prior multi-ethnic ALL GWAS (Blood 2019 133:724, *J. Natl. Cancer Inst* 2013 105:733).

The association tests in the replication series (for the 6 selected loci) already included genetic ancestry as co-variate. Therefore, the results are unlikely to be biased by population structure.

2. The replication statistics should adjust for multiple comparison e.g. Bonferroni correction would exclude the rs210143 (*BAK1*) from the “replicated” set. Much of the discussion and summary centers on this result that is marginal.

As the reviewer suggested, we estimated the false discovery rate (FDR) in the replication cohort using Benjamini Hochberg procedure. The FDR is <5% for the four SNPs with nominal p-value < 0.05 including rs210143 in the *BAK1* gene (**Page 12-13**). In addition, all four validated SNPs have the same risk alleles as in the discovery cohort.

Although we understand the concern of multiple testing, we respectfully argue that applying Bonferroni is not needed in this context given the low false discovery rate.

3. Conversely, the non-replication of 2 or 3 of the loci that met the GWAS criterion may be due to the assessment in a different ancestry (e.g. even absence of the risk allele). This should be discussed.

We acknowledge this point and have revised our text accordingly (**Page 8**). Specifically we state “Two of the genome-wide significant associations from our discovery meta-analysis were not replicated. This may be the consequence of a different population allelic structure between cohorts of different ancestry (Europeans in the discovery and non-Europeans in the replication) or population specific associations.”

4. It is not clear to this reviewer that all ten previously reported B-ALL risk loci are “confirmed” (page 4) rather than consistent with the previous reports. The question is whether the previous reported B-ALL risk loci were all identified in subjects independent of those used in the meta-analyses.

We agree that our original wording raises ambiguity and we have now revised accordingly (changed to “Meta-analysis identified 16 risk loci above genome-wide significance ($P < 5 \times 10^{-8}$) of which 10 are previously reported B-ALL risk loci”, **Page 4**).

5. There is a substantial amount of speculative discussion concerning the integration of GWAS signals with potential epigenetic mechanisms. The limitations of these in silico compilations should be emphasized including what appears to be a complete lack of any data that indicate that variants on susceptibility haplotypes are associated with the epigenetics.

We acknowledge this point and have revised our text (**Page 9**). Specifically, in the discussion, we now state: “Deciphering the functional consequences of risk loci is inherently challenging especially because these analyses are complicated by background haplotype structure. We have relied in part on integration of GWAS signals with in silico and publicly accessible epigenetic data, and the validity of these predictions requires experimental verification through laboratory assays in future studies.”

As explained above (Point 2, Reviewer 1), we have now also added new analyses of ALL cell line Hi-C data which provided further evidence for regulatory elements at these risk loci (**Supplementary Figure 8**).

Reviewer #3 (Remarks to the Author):

Vijayakrishnan et al present a meta-analysis of 4 GWAS datasets to identify novel risk variants for B-ALL. The cohort includes an impressive number of cases making it the largest B-ALL GWAS reported. They identify 4 novel risk loci, one for B-ALL in general, 2 specific for hyperdiploid ALL and 1 specific for ETV6-RUNX1 ALL. They go on to examine publically available epigenetic data from experiments using primary B cells and a B-lymphoblastoid cell line in an attempt to provide support for the likely relevance of these loci to B-ALL.

The authors have extensive experience in the field and the methods employed in the GWAS meta-analysis seem robust, but would require review from a statistician with expertise in this area. The manuscript is concise and clearly written. I am less convinced by the analysis of the epigenetic data, which fails to provide conclusive evidence for the importance of these SNPs in relation to gene expression in a definitively causative fashion.

Comments

1. The authors published a very similar paper earlier this year (Vijayakrishnan et al, Nat Comm 2019) which analyzed three of the 4 included GWAS datasets (the 2 UK studies and the German study), included 2,442 cases and identified 2 novel risk loci. This paper extends this analysis through the inclusion of the COG dataset adding a further 2,879 cases. I think the exact datasets included and the link to the recent paper should be made clearer in the introduction. While the paper does identify additional significant variants, and confirm their previous findings, the novelty is somewhat tempered by the previous paper.

To aid the reader we have linked text to specific references detailing each of the previous GWAS datasets in the Introduction (**Page 3**), including our recent B-ALL GWAS in Nat Comm in 2018 (there might have been a typo in this reviewer's comment when he/she referred to it as in 2019).

It should be noted that 1385 of the 2,879 cases in the COG_SJ cohort have never been included in our prior ALL susceptibility GWAS publications.

In terms of advancing the field we have not only doubled the number of ALL cases, we have also performed the first well-powered subtype-specific GWAS of B-cell ALL.

2. Figure 1. What do the asterisks refer to genes such as BAK1?

We have now removed asterisks in **Figure 1**, but have implanted a different colouring scheme to indicate new risk loci (as suggested by Reviewer 1).

3. It is noted that 4 of the 6 SNP were validated in the replication series (pg 4) with $P < 0.05$ – is this a significant P value at genome-wide significance, which was previously defined as $< 5 \times 10^{-8}$?

To improve clarity, we have now clearly stated that replication P-value was thresholded at 0.05 (**Page 12**).

4. Figure 2 and the associated legend require significant changes. The legend needs far more information to explain the lower part of each plot, and it takes a lot of work on behalf of the reader to work out what the authors are showing. Please explain why each sub-figure is split into 1) and 2). ATACseq and eQTL is not mentioned in the legend, nor the Y-axis for the eQTL section – does this P value include a correction for multiple testing (see Huang et al NAR 2018)? The ChromHMM data is not adequately explained and the same colors represent different things in different samples. Please order figures in relation to text; ie. 2A) 6p21, 2B) 5q31 etc.

We have extensively revised the legend for **Figure 2**. For each locus, plots are at 500Kb and a zoomed in version at 100kb resolution. This is now clearly stated.

5. The second half of the results, analyzing publically-available epigenetic data provides rather tenuous support to the identified loci in some cases, and most of these data are circumstantial, descriptive and correlative. While rs210143 at the 6p21 locus does appear to fall within a region of active promoter activity, supporting a causal role, 5q31 lies in an area of largely closed chromatin, so what is its putative role in altering gene expression in this setting? Do the SNPs alter predicted TF binding motifs?

Deciphering GWAS signaling is arguably challenging, as this reviewer alluded to. For this reason, leveraging publically accessible data to make predictions of functional effects of variants is routinely done in contemporary GWAS, and it has proven highly effective in identifying putative mechanisms and generating testable hypotheses. To address this concern, we have now added discussion to indicate the descriptive and correlative nature of these analyses and pointed out potential limitations (Page 9).

As we show in Figure 2 the 5q31 risk SNP is mapped to an enhancer region in two of the B-cells profiled.

6. Supp Figure 9 and 10 show that IGF2BP1 is overexpressed in ETV6-RUNX1 ALL. Do these cases carry the rs10853104 SNP and is it associated with increased IGF2BP1 expression?

To address this question, we have now analyzed a small cohort of 167 ALL cases (32 ETV6-RUNX1 ALL and 135 with other B-ALL subtypes) with both leukemia gene expression data and germline SNP genotype. As expected, 100% of the cases with ETV6-RUNX1 fusion showed dramatic over-expression of IGF2BP1 compared to other subtypes, but IGF2BP1 genotype was not associated the expression of this gene either within ETV6-RUNX1 ALL or in ALL cases negative for this fusion

We suspect that germline risk variant at this locus may only subtly affect IGF2BP1 expression. In cases with ETV6-RUNX1 fusion, the effect by SNP is simply masked by the dramatic overexpression resulted from somatic gene fusion. In cases without ETV6-RUNX1 fusion, IGF2BP1 expression is extremely low and we could not reliably quantify the effects of genotype.

We added discussion to address this (Page 8).

7. The authors mention that there was no statistically significant relationship between SNP genotype and outcome in the German data. Why was this analysis not extended to the combined dataset as outcome data are available on all these cohorts?

We now add outcome data from the COG_SJ series (Pages 7 and 16). Again, we did not observe any association with treatment outcome for these ALL risk variants.

8. Standard eQTL studies tend to link gene expression with SNP genotype, but do not necessarily take into account allelic expression. If the SNPs described truly alter gene expression, then one would expect allelic imbalance in gene expression (when SNP is heterozygous) - this is potentially detectable by analysis of coding heterozygous SNPs in the gene, or its 3'UTR. Similarly, TF binding would be allelically skewed. Have the authors looked into this?

With regard to allelic imbalance in gene expression: there is a requirement for the risk SNP is in strong LD ($r^2 > 0.8$) with a coding SNP. For all of the 4 novel risk loci SNPs there are no appropriate proxies.

With regard to allelic skewing of TF binding we would need to know the cell line is heterozygous, which we do not necessarily know and that there is no copy number variation at the locus which would artifactually skew the results. Furthermore, it would be highly advantageous to have control samples to address any inherent biases at the region. We agree that future studies are warranted to investigate these mechanistic details.

9. Given this study offers only a minor incremental advance from their recent Nat Comms paper, this study would have massively benefited from functional studies, for instance altering the SNPs with CRISPR in cell lines and analyzing gene expression to show true causation.

We agree with the reviewer that in-depth functional studies would help understand the exact mechanisms linking these variants to ALL, but also respectfully argue that such a body of work

(CRISPR of multiple SNPs in the LD regions of association) is outside the remit of the present publication.

In this one of the largest ALL GWAS ever reported, we 1) identified a multitude of novel risk loci for this cancer, 2) discovered subtype-specific loci for two of the most common subtypes of ALL, with highly plausible biological relevance (*IGF2BP1* and *BAK1*), 3) performed comprehensive analyses of ALL risk loci for TF binding patterns, 4) explored contribution of eQTL by the Mendelian randomization analysis, 5) conducted transcriptome genome-wide association study (TWAS), and 6) also modelled genetic heritability of ALL using the polygenic risk score method.

Taken together, we believe that this represents an expansive body of work which has advanced our understanding of ALL genetics in a significant and meaningful way.

We are grateful to the three reviewers' and having made the above changes hope our paper is now suitable for publication in *Nature Communications*.

Reviewers' comments:

Reviewer #1 (Remarks to the Author):

In the revised version of the manuscript, the authors have made a significant effort to address the points that were raised before, and the revised manuscript has improved significantly. Yet, mechanistic follow-up data on selected variants have not been provided. To address this point that was raised before, the authors have incorporated more datasets for correlative analyses. The resulting analysis indicates that differences in chromatin looping exist for some variants when normal, naïve B-cells are compared to B-ALL. While this is an interesting observation, it does not provide any mechanistic insight into how these variants could act. I would recommend to assemble the data in a way that these differences become more clearly visible, for example by integrating the Hi-C data from naïve B-cells vs. the NALM-6 data into the same figure.

Reviewer #2 (Remarks to the Author):

In general, the revised manuscript is substantially improved with clarifications in the presentation and some additional "functional" data. There are a few points that should be addressed.

1. Abstract: line 55. Is a bit strong considering the many caveats in interpretation and would strongly suggest changing "consistently points to" to "supports the role of" or "suggests" to strike a less definitive tone.
2. Results: line 109. It still may not be clear to the reader that these results are not an independent replication of the risk loci. This should be clarified here or at a minimum in a footnote to Table 1 and in the legend of Fig. 1.
3. Methods: line 328 and 329. The sentence needs to be reworded and details provided to understand what was done. At present it makes no sense -i.e. what is genomic inflation in the PCA? Also "carried out as appropriate" is not acceptable to this reviewer (i.e. what is appropriate?). My guess is that Eigen-vector co-variables were used in an effort to account/control for genomic inflation. You should state what was done, and at a minimum how many PCs were included and what was the residual genomic inflation factor was after these co-variables were included. Also, it is unclear whether the p values were adjusted for any residual lambda inflation after including ancestry co-variables.

Reviewer #3 (Remarks to the Author):

The authors have addressed the majority of points raised including clarification of patient cohorts and improvements to figure legends. The lack of functional characterisation of any of the identified SNPs means the paper remains descriptive and only a minor advance on their previous work, but is otherwise satisfactory. I suggest clarification of the B-cell lineage in the title.

We appreciate the new comments raised during the last round of review of our manuscript (NCOMMS-19-12126). We have now added further revision to address each of the questions/critiques as outlined below (marked in red). All changes are highlighted in yellow in the submitted manuscript files. For ease of evaluation, we have described our responses directly within the template of the reviewers' comments.

Reviewer #1:

In the revised version of the manuscript, the authors have made a significant effort to address the points that were raised before, and the revised manuscript has improved significantly. Yet, mechanistic follow-up data on selected variants have not been provided. To address this point that was raised before, the authors have incorporated more datasets for correlative analyses. The resulting analysis indicates that differences in chromatin looping exist for some variants when normal, naive B-cells are compared to B-ALL. While this is an interesting observation, it does not provide any mechanistic insight into how these variants could act.

I would recommend to assemble the data in a way that these differences become more clearly visible, for example by integrating the Hi-C data from naïve B-cells vs. the NALM-6 data into the same figure.

As suggested by this reviewer, we have now moved the CHiC track for Naïve B cells in **Fig. 2 to Supplementary Fig. 8**. As a result, now we have B cell CHiC data and B-ALL Hi-C data shown side-by-side in a single figure. We have revised the figure legends accordingly.

Reviewer #2:

In general, the revised manuscript is substantially improved with clarifications in the presentation and some additional "functional" data. There are a few points that should be addressed.

1. Abstract: line 55. Is a bit strong considering the many caveats in interpretation and would strongly suggest changing "consistently points to" to "supports the role of" or "suggests" to strike a less definitive tone.

As advised, we have changed to "suggests" in the Abstract (Page 2).

2. Results: line 109. It still may not be clear to the reader that these results are not an independent replication of the risk loci. This should be clarified here or at a minimum in a footnote to Table 1 and in the legend of Fig. 1.

As suggested, we have now added a clause ("Other risk loci were reported in previous GWAS using subsets of ALL cohorts included herein") in **Table 1** and in the legend of **Fig. 1** to clarify that this.

3. Methods: line 328 and 329. The sentence needs to be reworded and details provided to understand what was done. At present it makes no sense -i.e. what is genomic inflation in the PCA? Also "carried out as appropriate" is not acceptable to this reviewer (i.e. what is appropriate?). My guess is that Eigen-vector co-variates were used in an effort to account/control for genomic inflation. You should state what was done, and at a minimum how many PCs were included and what was the residual genomic inflation factor was after these co-variates were included. Also, it is unclear

whether the p values were adjusted for any residual lambda inflation after including ancestry covariates.

We apologize for the ambiguity and have added the details in the text, as follow: "To account for genomic inflation post imputation, top Eigenvectors from the principal component analysis were used as covariates in the final association analysis: the top two and five Eigenvectors for the UK_German GWAS and the COG_SJ GWAS, respectively. No further adjustments for P values were applied." Genomic inflation factor was reported as λ in **Supplementary Figure 2**.

Reviewer #3:

The authors have addressed the majority of points raised including clarification of patient cohorts and improvements to figure legends. The lack of functional characterisation of any of the identified SNPs means the paper remains descriptive and only a minor advance on their previous work, but is otherwise satisfactory. I suggest clarification of the B-cell lineage in the title.

As suggested, we have now changed the title to "Identification of four novel associations for B-cell acute lymphoblastic leukaemia risk".

REVIEWERS' COMMENTS:

Reviewer #2 (Remarks to the Author):

This revision of the manuscript has in this reviewer's opinion adequately addressed the remaining issues